# Sensitizing *Staphylococcus aureus* to antibacterial agents by decoding and blocking the lipid flippase MprF

Christoph J Slavetinsky[1,2,3,4,5]*, Janna N Hauser[1,3,4], Cordula Gekeler[1,3,4], Jessica Slavetinsky[1,3,4], André Geyer[1,3], Alexandra Kraus[6], Doris Heilingbrunner[6], Samuel Wagner[3,4,7], Michael Tesar[6], Bernhard Krismer[1,3,4], Sebastian Kuhn[1,3], Christoph M Ernst[1,3]*†, Andreas Peschel[1,3,4]

[1]Department of Infection Biology, Interfaculty Institute for Microbiology and Infection Medicine Tübingen (IMIT), Eberhard Karls University Tübingen, Tübingen, Germany; [2]Pediatric Gastroenterology and Hepatology, University Children's Hospital Tübingen, Eberhard Karls University Tübingen, Tübingen, Germany; [3]German Centre for Infection Research (DZIF), Partner Site Tübingen, Tübingen, Germany; [4]Cluster of Excellence "Controlling Microbes to Fight Infections," University of Tübingen, Tübingen, Germany; [5]Pediatric Surgery and Urology, University Children's Hospital Tübingen, Eberhard Karls University Tübingen, Tübingen, Germany; [6]MorphoSys AG, Planegg, Germany; [7]Section of Cellular and Molecular Microbiology, Interfaculty Institute for Microbiology and Infection Medicine Tübingen (IMIT), University of Tübingen, Tübingen, Germany

*For correspondence:
christoph.slavetinsky@med.uni-tuebingen.de (CJS);
cmernst@broadinstitute.org (CME)

Present address: †Christoph M. Ernst, Department of Molecular Biology and Center for Computational and Integrative Biology, Massachusetts General Hospital, Harvard Medical School, Boston, United States

**Abstract** The pandemic of antibiotic resistance represents a major human health threat demanding new antimicrobial strategies. Multiple peptide resistance factor (MprF) is the synthase and flippase of the phospholipid lysyl-phosphatidylglycerol that increases virulence and resistance of methicillin-resistant *Staphylococcus aureus* (MRSA) and other pathogens to cationic host defense peptides and antibiotics. With the aim to design MprF inhibitors that could sensitize MRSA to antimicrobial agents and support the clearance of staphylococcal infections with minimal selection pressure, we developed MprF-targeting monoclonal antibodies, which bound and blocked the MprF flippase subunit. Antibody M-C7.1 targeted a specific loop in the flippase domain that proved to be exposed at both sides of the bacterial membrane, thereby enhancing the mechanistic understanding of bacterial lipid translocation. M-C7.1 rendered MRSA susceptible to host antimicrobial peptides and antibiotics such as daptomycin, and it impaired MRSA survival in human phagocytes. Thus, MprF inhibitors are recommended for new antivirulence approaches against MRSA and other bacterial pathogens.

## Editor's evaluation

This study uses an innovative anti-virulence approach based on monoclonal antibodies that target the *Staphylococcus aureus* lipid flippase involved in tolerance to cationic peptides. The authors show that this strategy re-sensitizes antibiotic-resistant *S. aureus* and serves as a proof of principle for anti-virulence approaches to target bacterial infections.

## Introduction

The continuous increase of antibiotic resistance rates undermines the significance and efficacy of available antibiotics against bacterial infections (*Årdal et al., 2020*). Several opportunistic antibiotic-resistant bacterial pathogens including methicillin-resistant *Staphylococcus aureus* (MRSA), vancomycin-resistant enterococci, and extended-spectrum beta-lactam or carbapenem-resistant proteobacteria impose a continuously growing pressure on modern healthcare systems (*Tacconelli et al., 2018*). MRSA is responsible for a large percentage of superficial and severe bacterial infections and the available last-resort antibiotics are much less effective than beta-lactams (*Lee et al., 2018*). Unfortunately, no new class of antibiotics has entered the clinical phase since the introduction of the lipopeptide antibiotic daptomycin in 2003 (*Årdal et al., 2020*). Novel anti-infective strategies that would circumvent on the one hand the difficulties in identifying new microbiota-preserving small-molecule antimicrobials and, on the other hand, the enormous selection pressures exerted by broad-spectrum antibiotics, are discussed as potential solutions against a looming postantibiotic era (*Dickey et al., 2017*). Such strategies could be based for instance on therapeutic antibodies or bacteriophages, which usually have only a narrow activity spectrum. A possible direction could be the inhibition of bacterial targets that are of viable importance only during infection (*Lakemeyer et al., 2018*). Blocking such targets by so-called antivirulence or antifitness drugs would preserve microbiome integrity and create selection pressure for resistance-conferring mutations only on invading pathogens. Interfering with bacterial virulence factors should ameliorate the course of infection and enable more effective bacterial clearance by the immune system or by antibiotics.

Monoclonal antibodies (mABs) directed against antivirulence targets could be interesting alternatives provided the target can be reached by comparatively large antibody molecules. Therapeutic mABs are used in several malignant, inflammatory, and viral diseases (*Qu et al., 2018*; *O'Brien et al., 2021*) and have proven efficacy in toxin-mediated bacterial infections such as anthrax or clostridial toxin-mediated diseases (*Dickey et al., 2017*; *Migone et al., 2009*; *Lowy et al., 2010*). Apart from toxin neutralization, however, mABs have hardly been applied in antimicrobial development programs. Moreover, in-depth molecular studies are necessary to devise most promising targets for mABs and elucidate if and how mAB binding could disable pathogens to colonize and infect humans.

The multiple peptide resistance factor (MprF), a large integral membrane protein, is crucial for the capacity of bacterial pathogens such as *S. aureus* to resist cationic antimicrobial peptides (CAMPs) of the innate immune system and CAMP-like antibiotics such as daptomycin (*Peschel et al., 2001*; *Ernst and Peschel, 2011*; *Slavetinsky et al., 2017*). MprF is highly conserved and can be found in various Gram-positive or Gram-negative pathogens (*Slavetinsky et al., 2017*). MprF proteins proved to be crucial for in vivo virulence of various pathogens in infection models (*Peschel et al., 2001*; *Thedieck et al., 2006*; *Maloney et al., 2009*) and when exposed to human phagocytes as a result of increased resistance to phagocyte-derived antimicrobial agents such as CAMPs (*Slavetinsky et al., 2017*; *Kristian et al., 2003*). Some parts of the protein are located at the outer surface of the cytoplasmic membrane and could in principle be reached by mABs (*Ernst et al., 2009*; *Ernst et al., 2015*). MprF forms oligomers and it is a bifunctional enzyme, which can be separated into two distinct domains (*Ernst et al., 2015*). The C-terminal domain synthesizes positively charged lysyl-phosphatidylglycerol (LysPG) from a negatively charged phosphatidylglycerol (PG) acceptor and a Lys-tRNA donor substrate, while the N-terminal domain translocates newly synthesized LysPG from the inner to the outer leaflet of the cytoplasmic membrane and thus functions as a phospholipid flippase (*Ernst et al., 2009*; *Roy and Ibba, 2009*). The exposure of LysPG at the outer surface of the membrane reduces the affinity for CAMPs and other antimicrobials (*Ernst et al., 2009*). Notably, *mprF* is a major hot spot for gain-of-function point mutations that lead to daptomycin resistance, acquired during therapy of *S. aureus* infections (*Ernst et al., 2018*).

In order to assess the suitability of MprF as a target for antivirulence agents we developed mABs targeting several epitopes of potential extracellular loops of its transmembrane part and analyzed their capacity to bind specifically to *S. aureus* MprF. We identified a collection of mABs, which did not only bind to but also inhibited the LysPG flippase domain of MprF. Our results suggest that a specific loop between two of the transmembrane segments (TMSs) of MprF is exposed at both sides of the membrane suggesting an unusual, potentially flexible topology of this protein part, which may be involved in LysPG translocation. Accordingly, targeting this loop with a specific mAB inhibited the MprF flippase function, rendered *S. aureus* susceptible to killing by antimicrobial host peptides and

daptomycin, and reduced *S. aureus* survival when challenged by human CAMP-producing polymorphonuclear leukocytes (PMNs).

## Results

### Generation of mABs binding to putative extracellular loops of MprF

The hydrophobic part of *S. aureus* MprF appears to include 14 TMS connected by loops with predicted lengths between 2 and 56 amino acids (*Ernst et al., 2015*). Several of the loops are located at the outer surface of the cytoplasmic membrane, accessible to mABs (*Figure 1A*). Peptides representing 4 loops with a minimum length of 13 amino acids were synthesized with N- and C-terminal cysteine residues to allow cyclization (*Supplementary file 1a*). The peptides corresponded to three loops predicted to be at the outer membrane surface (loops 1, 9, and 13) and loop 7, the location of which has remained ambiguous due to conflicting computational and experimental findings (*Ernst et al., 2015*). The N-terminal amino groups of cyclized peptides were linked to biotin to facilitate their recovery and detection. The antigen peptides were incubated with MorphoSys's Human Combinatorial Antibody Library (HuCAL), a phage display library expressing human Fab fragments with highly diverse variable regions at the phage surface (*Prassler et al., 2011*). Antigen-binding phages were enriched in three iterative rounds of panning in solution and antigen-phage complexes were captured with streptavidin-coated beads. The bound phages were extensively washed to remove unspecifically binding phages, eluted, and propagated in *Escherichia coli* for a subsequent panning round. Washing steps were prolonged and antigen concentrations reduced from round one to round three to increase stringency and discard antibodies with low specificity and affinity. DNA of the eluted, antigen-specific phages was isolated and subcloned in specific *E. coli* expression vectors to yield His-tagged fragment antigen-binding (Fab) molecules. 368 individual colonies per antigen were picked and Fab fragments were expressed and purified. A representative selection of 24 unique Fabs against all 4 peptides were converted to human IgG by cloning in an IgG1 expression vector system and expression in human HKB11 cells and IgGs were purified via protein A chromatography, as recently described (*Prassler et al., 2011*) (see graphical workflow in *Figure 1—figure supplement 1*).

The IgGs were analyzed for binding to the corresponding antigen peptides and also to the three noncognate peptides to assess their selectivity, by ELISA with streptavidin-coated microtiter plates (*Figure 1—figure supplement 2*). Peptide one bound IgGs developed against different antigen peptides indicating that it may bind antibodies with only low selectivity. Antibodies directed against the four targeted MprF loops were selected based on affinity and analyzed for binding to *S. aureus* SA113 cells expressing or not expressing MprF. Both, the 'wild-type' and *mprF* deletion mutant strains lacked the gene for protein A (*spa*), which would otherwise unspecifically bind to IgG (*Kim et al., 2012*). Bacteria were adsorbed to microtiter plates, blocked with bovine serum albumin, and incubated with IgGs, which were then detected with goat antihuman IgG after extensive washing. Antibodies M-C1, M-C7.1, M-C7.3, M-C9.1, M-C13.1, and M-C13.2 bound significantly stronger to MprF-expressing *S. aureus* (SA113Δ*spa*) compared to MprF-deficient *S. aureus* (SA113Δ*spa*Δ*mprF*), while the humanized isotype control mAB L-1 showed no specific binding (*Figure 1B*). An additional mAB directed against the *S. aureus* surface protein EbpS served as positive control, showing equal affinity toward SA113Δ*spa* and SA113Δ*spa*Δ*mprF* (*Figure 1B*). These findings are in agreement with the location of the loops 1, 9, and 13 at the outer surface of the cytoplasmic membrane, confirming the overall topology of MprF (*Figure 1A*). Of note, loop 7 between potential TMS 7 and 8 whose location had previously remained controversial was detected by antibodies M-C7.1 and M-C7.3 (*Figure 1B*) indicating that loop 7 is accessible from the outside. Antibodies M-C1, M-C7.1, M-C9.1, and M-C13.1 showed the strongest binding to MprF.

### The MprF epitope bound by M-C7.1 is located at both, outer, and inner surface of the cytoplasmic membrane

The MprF loop 7 between TMS 7 and 8 bound by M-C7.1 seemed to have an ambiguous position within the cytoplasmic membrane because the flanking TMS have a comparatively low content of hydrophobic amino acids. Accordingly, only some topology analysis algorithms predict its location at the outer surface of the cytoplasmic membrane and a previous experimental topology investigation using a set of translational fusions with enzymes that are active only at intracellular or extracellular

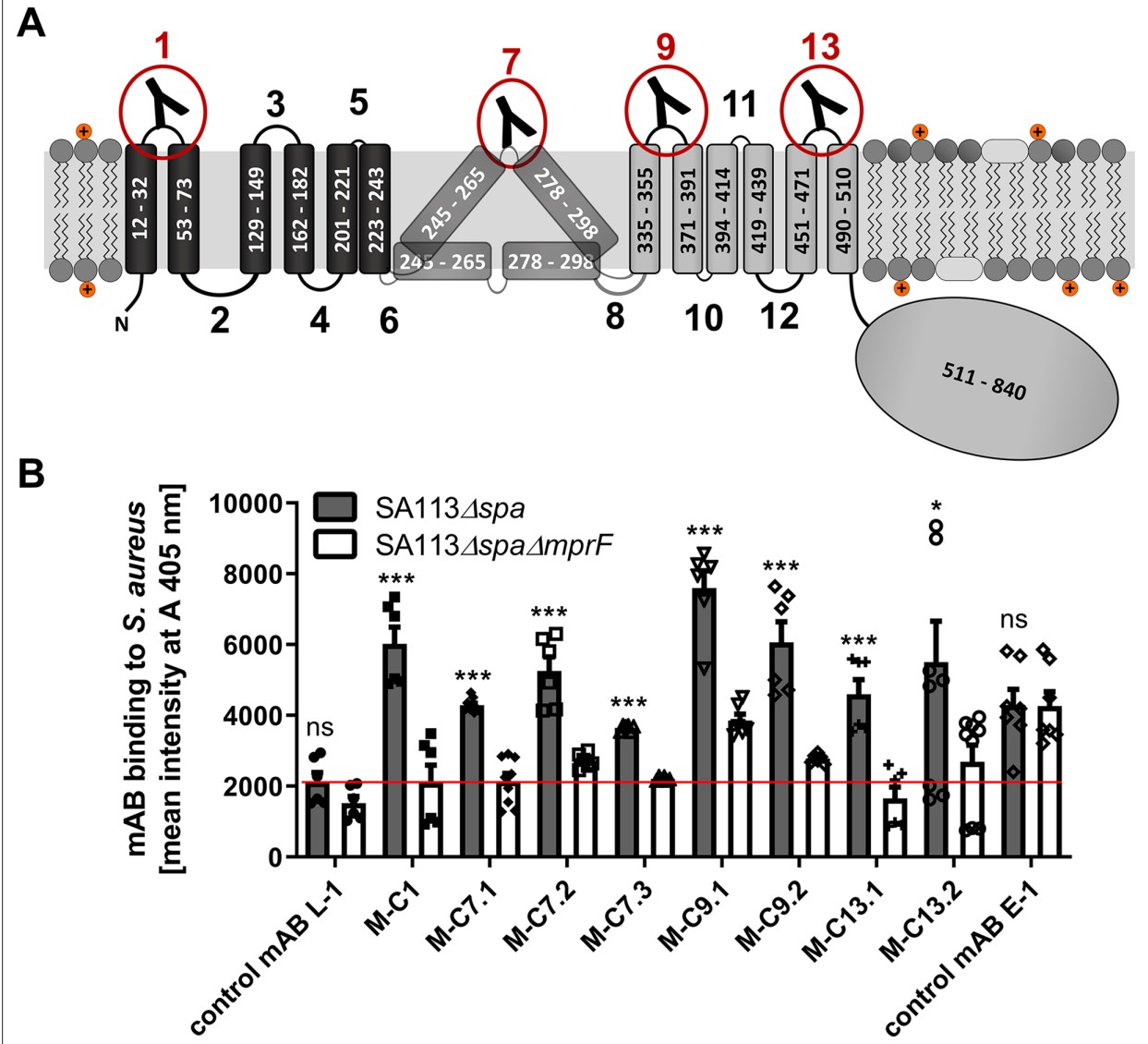

**Figure 1.** Multiple peptide resistance factor (MprF) topology and binding of monoclonal antibodies (mABs) to MprF-expressing *S. aureus* cells. (**A**) MprF membrane topology is given according to our recent study (***Ernst et al., 2015***) showing synthase and flippase domains in gray and black, respectively. Amino acid (aa) positions of transmembrane segment (TMS) and the C-terminal hydrophilic domain are indicated. TMSs from aa 245–265 and aa 278–298 are shown in two alternative positions as computational and experimental results of transmembrane topology have been contradictory (***Ernst et al., 2015***). Localizations of MprF's TMS-connecting loops are numbered starting from the N-terminus, antibody-targeted loops are indicated by red circles and antibody symbols. (**B**) Specific binding of mABs (100 nM) to *S. aureus* was analyzed by ELISA using SA113 strains deficient in the IgG-binding protein A (Spa) comparing the SA113 *spa* mutant (*Δspa*) and *spa mprF* double knockout mutant (*ΔspaΔmprF*). The red line indicates the mean intensity measured at A 405 nm (affinity) of the isotype control mAB L-1 bound to *S. aureus* SA113*Δspa*. Means and standard error of the mean (SEM) of at least three biological replicates are shown. Significant differences between SA113*Δspa* and SA113*ΔspaΔmprF* were calculated by Student's paired *t*-test (ns, not significant; *p < 0.05; ***p < 0.0001).

The online version of this article includes the following figure supplement(s) for figure 1:

**Figure supplement 1.** Workflow for the development of multiple peptide resistance factor (MprF)-specific monoclonal antibodies (mABs).

**Figure supplement 2.** Specific binding of selected monoclonal antibodies (mABs) to cyclic multiple peptide resistance factor (MprF)-derived target peptides analyzed by ELISA.

location has revealed a preferential location at the inner cytoplasmic membrane surface in *E. coli* (**Ernst et al., 2015**). Since our M-C7.1- and M-C7.3-binding experiments indicated accessibility of loop 7 from the outside, we revisited its location in *S. aureus* with two experimental strategies.

To confirm that M-C7.1 has access to its cognate MprF antigen epitope at the outer surface of the cytoplasmic membrane in intact bacterial cells, *S. aureus* SA113Δ*spa*Δ*mprF*-expressing MprF (pRB-MprF) was grown in the presence of M-C7.1. Cells were then disrupted, membranes were solubilized by treatment with the mild nonionic detergent *n*-dodecyl-β-D-maltoside (DDM), and proteins were separated in nondenaturing PAGE gels (BN-PAGE). If M-C7.1 bound before cell disruption and remained tightly attached to MprF, it should shift the MprF bands in Western blots of the blue native gels and the MprF–M-C7.1 complex should be detectable after Western blotting with a human IgG-specific secondary antibody. In order to detect MprF independently of M-C7.1, an MprF variant translationally fused to green-fluorescent protein (MprF-GFP, expressed from plasmid pRB-MprF-GFP), which is detectable by a GFP-specific primary antibody (**Ernst et al., 2015**), was also used and treated in the same way. To detect potentially nonshifted MprF proteins in the native MprF variant (pRB-MprF), M-C7.1 was used as additional primary antibody. *S. aureus* SA113Δ*spa*Δ*mprF* bearing the empty vector (pRB) was used as a negative control. In addition to M-C7.1, all three *S. aureus* strains were also incubated with the control antibody L-1. *S. aureus* cells expressing either unmodified MprF or MprF-GFP yielded a protein band migrating at a molecular weight of around 900 kDa when preincubated with M-C7.1 but not with L-1 or in cells with the empty-vector control (pRB) (**Figure 2A**; **Figure 2—figure supplement 2**). In contrast, the empty-vector control (pRB) strain showed an unspecific band at 300 kDa when preincubated with M-C7.1 or at 150 kDa when preincubated with L-1 but no MprF-specific band (**Figure 2A**; **Figure 2—figure supplement 2**). The 900 kDa band of MprF-GFP was detected by both, M-C7.1 and, with a weaker signal, anti-GFP, confirming the identity of MprF (**Figure 2A**; **Figure 2—figure supplement 2**). We could recently show that MprF forms oligomers in the staphylococcal membrane, which were migrating at ca. 300 and 600 kDa (**Ernst et al., 2015**). Bands migrating at similar heights (ca. 250 and 500 kDa) were detected specifically in the MprF-GFP lanes after preincubation with either M-C7.1 or L-1 (**Figure 2A**; **Figure 2—figure supplement 2**) suggesting that they represent the oligomerized but not the mAB-complexed MprF proteins (**Ernst et al., 2015**). Therefore, the 900 kDa bands probably represent a complex formed by MprF and M-C7.1, which confirms that M-C7.1 specifically binds MprF loop seven in live *S. aureus* cells. The fact, that M-C7.1 used as primary antibody was not able to detect 250 and 500 kDa bands of noncomplexed MprF proteins while the MprF–M-C7.1 complex can directly be detected via anti-human secondary antibody, suggests that M-C7.1 binding only occurs in the living staphylococcal cell. Of note, the MprF–M-C7.1 complex migrating at around 900 kDa indicates that higher-order MprF multimers were shifted by complex formation with M-C7.1.

The position of MprF loop 7 was further investigated by inserting a cysteine residue into the loop and analyzing the capacity of the membrane-impermeable agent Na*-(3-maleimidylpropionyl)-biocytin (MPB) to label cysteines covalently, using the substituted cysteine accessibility method (SCAM) (**Bogdanov et al., 2005**). MPB treatment of intact *S. aureus* cells should only lead to labeling of extracellular protein portions while blocking cysteines at the outside with 4-acetamido-4'-maleimidylstilbene-2,2'-disulfonic acid (AMS), followed by cell homogenization and addition of MPB should allow to label only protein parts at the inner surface of the membrane according to a previously established method in *E. coli* (**Bogdanov et al., 2005**). An MprF variant lacking all native cysteine residues was generated to exclude background signals. Cysteine residues in MprF were exchanged against serine or alanine residues to minimize structural or functional changes. Native *S. aureus* MprF contains six cysteines none of which is conserved in MprF proteins from other bacteria suggesting that they do not have critical functions. The cysteine-deficient mutant protein was found to be indeed functional because it decreased the susceptibility of the *S. aureus mprF* mutant (SA113Δ*mprF*) to daptomycin (**Figure 2—figure supplement 1**), which depends on intact MprF synthase and flippase activities (**Ernst et al., 2009**). However, the mutated proteins conferred lower resistance levels than the wild-type protein, presumably because of less efficient protein folding or stability. Cysteines were then inserted into MprF loop seven (T263) and, as a control, into the first intracellular loop (loop 2, A99) and the last extracellular loop (loop 13, T480), the localization of which had been consistently confirmed by previous computational and experimental analyses (**Ernst et al., 2015**; **Figures 1A and 2C**). Amino acids for exchange were chosen according to a predicted weak effect for functional

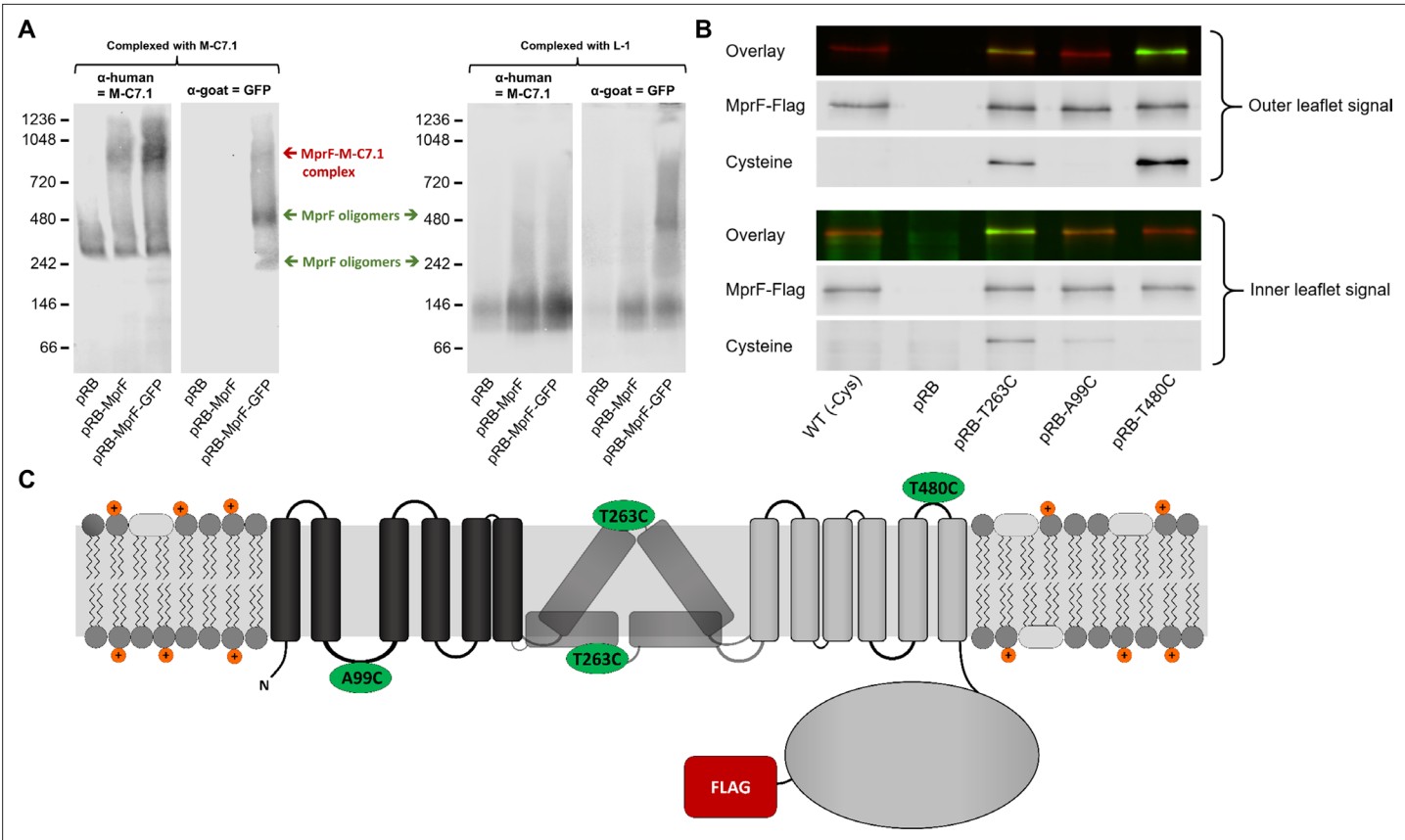

**Figure 2.** Binding of M-C7.1 to multiple peptide resistance factor (MprF) and membrane localization of the M-C7.1-targeted MprF loop 7. (**A**) Detection of M-C7.1 binding to MprF. Plasmid-encoded native and green fluorescent protein(GFP)-tagged MprF variants were expressed in *S. aureus* SA113Δ*spa*Δ*mprF* and living cells were preincubated with M-C7.1 or the isotype control monoclonal antibody (mAB) L-1 (in order to form MprF–mAB complexes). MprF variants complexed with M-C7.1 or control mAB L-1, respectively, were detected by blue native PAGE followed by Western blotting using two different primary (anti-GFP or M-C7.1) and corresponding secondary antibodies. SA113Δ*spa*Δ*mprF* expressing the empty vector (pRB) served as negative control. Molecular masses in kDa of marker proteins are given on the left of the blot. Arrows mark both the MprF–M-C7.1 complex at 900 kDa and the MprF oligomers at 250 and 500 kDa, which were previously described (***Ernst et al., 2015***). (**B**) Cellular localization of the antigen epitope of M-C7.1 using the substituted cysteine accessibility method (SCAM) for specific loops between the MprF transmembrane segments (TMSs). The substituted cysteine T263C is localized in M-C7.1's target peptide sequence in MprF. Substitution of A99C served as inside control, substitution of T480C served as outside control (see topology model, part C). *S. aureus* SA113Δ*mprF* expressing the empty vector (pRB) and an MprF variant lacking all native cysteines (wild-type [WT] (-Cys)) served as additional negative controls. All MprF variants were plasmid-encoded, FLAG tagged at the C-terminus to allow immunoprecipitation and detection, and were expressed in *S. aureus* SA113Δ*mprF*. Substituted extracellular cysteine residues were labeled with Na*-(3-maleimidylpropionyl)-biocytin (MPB) (outer leaflet signal, green in overlay), while labeling of substituted internal cysteine with MPB was performed after the blocking of external cysteines with 4-acetamino-4'-maleimidylstilbene-2,2'-disulfonic acid (AMS) (inner leaflet signal, green in overlay). MprF was detected via antibody staining by an anti-FLAG antibody (red in overlay). (**C**) MprF topology showing location and amino acid exchanges of artificial cysteine residues for SCAM detection in green.

The online version of this article includes the following figure supplement(s) for figure 2:

**Figure supplement 1.** Effects of cysteine replacement and insertion on multiple peptide resistance factor (MprF) function, assessed by measuring daptomycin susceptibility.

**Figure supplement 2.** Detection of M-C7.1 binding to multiple peptide resistance factor (MprF).

changes by respective substitution with cysteine using https://predictprotein.org (***Yachdav et al., 2014***). The prediction is based on a machine learning program integrating both evolutionary information and structural features such as predicted secondary structure and solvent accessibility to evaluate the effect of amino acid exchanges in a protein sequence (***Yachdav et al., 2014***). The SCAM approach further confirmed the topology of intracellular loop 2 and extracellular loop 13 (***Figure 2B***), thereby demonstrating that the technique can lead to reliable results in *S. aureus*. Notably, MprF loop 7 was found in both locations, at the inner and outer surface of the membrane. This finding corroborates the

ambiguous position of loop 7 and suggests that this loop may have some degree of mobility in the membrane, which may reflect the lipid translocation process. The finding also clarifies why M-C7.1 has the capacity to bind MprF loop 7 at the outer surface of the cytoplasmic membrane.

## mABs binding to putative extracellular loops of MprF render *S. aureus* susceptible to CAMPs

MprF confers resistance to cationic antimicrobials such as the bacteriocin nisin by reducing the negative net charge of the membrane outer surface (*Peschel et al., 2001*; *Ernst et al., 2009*). If the mABs would not only bind to but also inhibit the function of MprF, *S. aureus* should become more susceptible to nisin. The six mABs with confirmed specific binding to MprF and the isotype control mAB L-1 were analyzed for their capacity to increase the susceptibility of *S. aureus* SA113 to nisin. For those initial screening experiments protein A (*spa*) mutants were used to diminish effects of unspecific IgG binding to protein A. Bacteria were grown in the presence of one of the mABs and then incubated with nisin at the IC$_{50}$ followed by quantification of viable bacterial cells. Two of the MprF-specific mABs targeting loops 7 or 13 increased the sensitivity of *S. aureus* SA113Δ*spa* to nisin while the other mABs and the isotype control mAB L-1 had no significant impact (*Figure 3A*). mAB M-C7.1 caused the strongest sensitization (*Figure 3A*) and it synergized with nisin in a dose-dependent fashion (*Figure 3—figure supplement 1*). It was therefore selected for further analysis using the highly prevalent and virulent community-associated methicillin-resistant *S. aureus* (CA-MRSA) USA300 clone (*Otto, 2013*).

For all following experiments *S. aureus* wild-type (WT) strains with intact protein A were used to make sure that the observed sensitization to CAMPs was not affected by unspecific antibody binding to protein A.

M-C7.1 was found to also increase the susceptibility of USA300 WT to the human CAMP LL-37, a host defense peptide produced by epithelial and phagocyte cells (*Pinheiro da Silva and Machado, 2017*), and to daptomycin, a lipopeptide antibiotic in clinical use sharing physicochemical and antibacterial properties with CAMPs (*Bayer et al., 2013*), in addition to nisin (*Figure 3B–D*). M-C7.1 also increased the antimicrobial activity of daptomycin against the daptomycin-resistant (DAP-R) clinical CA-MRSA isolate 703 possessing the gain-of-function mutation S295L in MprF (*Jones et al., 2008*; *Figure 3E*). Of note, M-C7.1 could reduce daptomycin minimal inhibitory concentration (MIC) of both, *S. aureus* SA113 WT and the DAP-R strain 703 (*Figure 3F*), suggesting that M-C7.1 may potentially be able to overcome daptomycin resistance during therapy. M-C7.1 but not the isotype control mAB L-1 was able to inhibit growth of USA300 WT in the presence of subinhibitory concentrations of nisin (*Figure 3G*). Thus, mABs specific for certain extracellular loops of MprF may not only bind to MprF but also inhibit its function. When USA300 WT was passaged for several days through media with M-C7.1 at 10 or 100 nM and with or without subinhibitory daptomycin (0.5 µg/ml), no point mutations in *mprF* were found suggesting that the MprF segment targeted by M-C7.1 is not prone to quickly occurring escape mutations.

## M-C7.1 inhibits the flippase function of MprF

M-C7.1 binds MprF at the junction between the LysPG synthase and flippase domains (*Figure 1*). The lipid patterns of SA113 WT treated with M-C7.1 at concentrations that increased the susceptibility to nisin or with the isotype control mAB L-1 were compared but showed no differences indicating that the synthase function of MprF was not inhibited by M-C7.1 (*Figure 4A*). The flippase activity of MprF promotes the exposure of positively charged LysPG at the outer surface of the cytoplasmic membrane thereby reducing the affinity for the small cationic protein cytochrome C (*Peschel et al., 1999*), which binds preferentially to negatively charged PG, or for calcium-bound annexin V (*Yount et al., 2009*). These model proteins have been shown to allow a sensitive assessment of changes in the surface charge of *S. aureus* in several previous studies (*Ernst et al., 2009*; *Ernst et al., 2018*; *Slavetinsky et al., 2012*). Treatment of SA113 WT with M-C7.1 at concentrations that increased the susceptibility to nisin led to a significant increase in the capacity to bind cytochrome C or annexin V compared to SA113 WT without treatment or treated with isotype control mAB L-1 (*Figure 4B, C*) thereby indicating that M-C7.1 inhibits the flippase function of MprF. It is tempting to speculate that the protein region of MprF loop 7 and adjacent TMSs may accomplish a crucial function in the process of phospholipid translocation, as suggested by the dynamic localization of loop 7 on the cytoplasmic and external faces of the membrane.

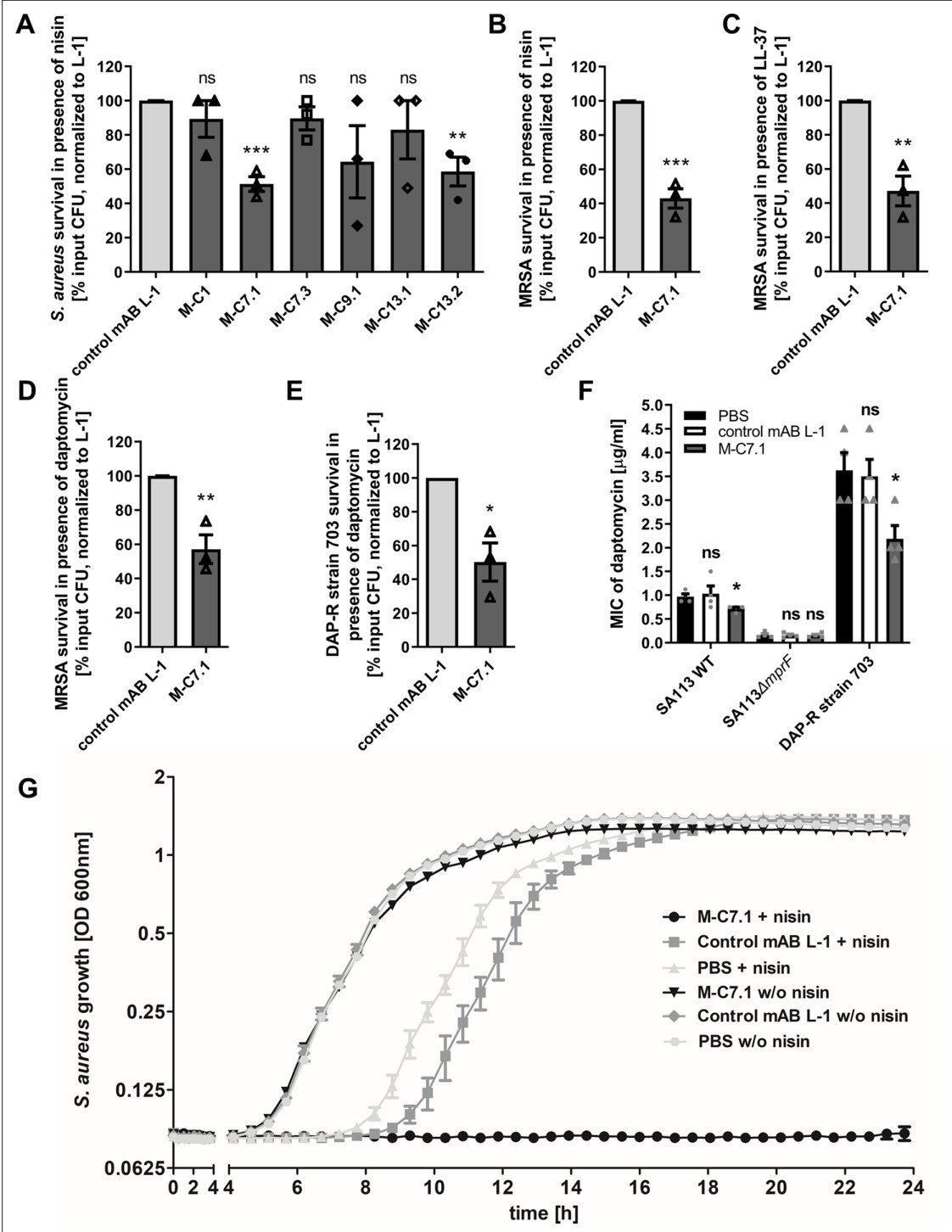

**Figure 3.** Killing and growth inhibition of *S. aureus* by antimicrobial peptides and antibiotics in the presence of M-C7.1.
(**A**) Survival of *S. aureus* SA113Δ*spa* in the presence of nisin and 100 µg/ml anti-multiple peptide resistance factor (MprF) antibodies compared to 100 µg/ml control monoclonal antibody (mAB) L-1. Surviving colony-forming units (CFU) of *S. aureus* incubated with one of the antibodies and nisin were analyzed after 2 hr incubation and the negative control (isotype mAB L-1) was set to 100% survival. (**B**) Survival of community-associated methicillin-resistant *S. aureus* (CA-MRSA) wild-type (WT) strain USA300 in the presence of nisin and 100 µg/ml M-C7.1 compared to the isotype control mAB L-1. (**C**) Survival of USA300 WT in the presence of LL-37 and 100 µg/ml M-C7.1 compared to the isotype control mAB L-1. (**D**) Survival of USA300 WT in the presence of daptomycin and 100 µg/ml M-C7.1 compared to the isotype control mAB L-1. (**E**) Survival of daptomycin-resistant (DAP-R) CA-MRSA strain 703 (*Jones et al., 2008*) in the presence of daptomycin and 100 µg/ml M-C7.1 compared to the isotype control mAB L-1. (**F**) Daptomycin MICs of SA113 WT, SA113Δ*mprF* and DAP-R strain 703 when pretreated with PBS compared to 100 µg/ml control mAB L-1 and to 100 µg/ml M-C7.1. (**G**)

*Figure 3 continued on next page*

*Figure 3 continued*

Growth inhibition of USA300 WT in the presence of 4 µg/ml nisin and 1 µM M-C7.1 compared to 1 µM isotype control mAB L-1. Wells without nisin and/ or antibodies served as additional negative controls. The means + standard error of the mean (SEM) of results from at least three biological replicates are shown in (A)–(F). (G) shows means + SEM of technical triplicates from a representative experiment of three biological replicates. Values for M-C7.1 or other anti-MprF antibodies that were significantly different from those for the isotype control mAB L-1 in (A)–(F), calculated by Student's paired *t*-test are indicated (\*p < 0.05; \*\*p < 0.01; \*\*\*p < 0.0001).

The online version of this article includes the following figure supplement(s) for figure 3:

**Figure supplement 1.** Dose-dependent support of M-C7.1 of *S. aureus* killing by nisin.

## M-C7.1 treatment abrogates *S. aureus* survival in phagocytes

The capacity of PMNs to kill phagocytosed bacteria does not only rely on the oxidative burst but also on the activity of LL-37 and other CAMPs and antimicrobial proteins (*Spaan et al., 2013*). Accordingly, *S. aureus mprF* mutants are more susceptible to PMN killing than the parental strains while their opsonization and phagocytosis by PMNs remains unaltered (*Peschel et al., 2001*; *Kristian et al., 2003*). The increased susceptibility of M-C7.1-treated *S. aureus* to CAMPs should therefore alter its survival ability in PMNs. CA-MRSA WT strain USA300 was pretreated with mABs, opsonized with normal human serum, and exposed to human PMNs. Treatment with M-C7.1 or the isotype control mAB L-1 did not alter the rate of PMN phagocytosis, but M-C7.1-treated USA300 cells were significantly more rapidly killed by PMNs than those treated with the isotype control antibody (*Figure 5*). This finding reflects our previous reports on reduced survival of MprF-deficient *S. aureus* in PMNs (*Peschel et al., 2001*; *Kristian et al., 2003*). Thus, *S. aureus* treatment with M-C7.1 might reduce the capacity to persist in infections and may help to blunt the virulence of *S. aureus* in invasive infections.

## Discussion

Monoclonal therapeutic antibodies have proven efficacy for neutralization of bacterial toxins such as *Clostridium botulinum* or *Clostridioides difficile* toxins, and mAB-based therapies targeting the *S. aureus* alpha toxins and leukotoxins are currently developed (*Dickey et al., 2017*; *Tabor et al., 2016*). Moreover, therapeutic mABs binding to *S. aureus* surface molecules to promote opsonic phagocytosis have been assessed in preclinical and clinical trials (*Missiakas and Schneewind, 2016*). In contrast, mABs inhibiting crucial cellular mechanisms of bacterial pathogens have hardly been assessed so far (*Zheng et al., 2019*). We developed specific mABs, which can block the activity of *S. aureus* MprF, the first described bacterial phospholipid flippase. Our mABs did not mediate increased internalization of *S. aureus* cells, most probably because the bacterial cell wall is too thick to allow binding of the mAB FC part to phagocyte FC receptors. However, the inhibition of MprF by M-C7.1 sensitized *S. aureus* to CAMPs and daptomycin and promoted killing by human PMNs, which use CAMPs as an important component of their antimicrobial arsenal. Specific inhibition of MprF could therefore promote the capacity of human host defense to clear or prevent a *S. aureus* infection and would, at the same time, increase the susceptibility of *S. aureus* to CAMP-like antibiotics such as daptomycin. Thus, targeting bacterial defense mechanisms provides a promising concept for antivirulence therapy. The capacity of antibodies to traverse the cell wall and reach the cytoplasmic membrane of *S. aureus* has remained controversial (*Reichmann et al., 2014*). However, other labs' and our findings demonstrate that specific antibodies can reach membrane-associated epitopes in *S. aureus* (*Mishra et al., 2012*; *Weisman et al., 2009*). A recent atomic-force microscopy-based study revealed large, irregular pores in the cell wall of *S. aureus*, which might allow large proteins such as antibodies to reach the cytoplasmic membrane (*Pasquina-Lemonche et al., 2020*). M-C7.1 sensitized *S. aureus* also in the presence of protein A indicating that the unspecific IgG binding by protein A does not interfere with the M-C7.1 capacity to block MprF.

Specific binding of MprF antibody M-C7.1 to the extracellular loop 7 inhibited the flippase function of MprF, which indicates that this loop plays a crucial role in the lipid translocation process. Loop 7 is located between predicted TMS 7 and 8, and its presence at the outer surface of the cytoplasmic membrane had remained elusive since computational and experimental analyses had yielded conflicting results (*Ernst et al., 2015*). Surprisingly, we found loop 7 to be accessible from both, the outside and inside of the cytoplasmic membrane. This suggests that the protein part formed by loop 7 and adjacent TMS 7 and 8 may change its position in the protein complex, moving between the two

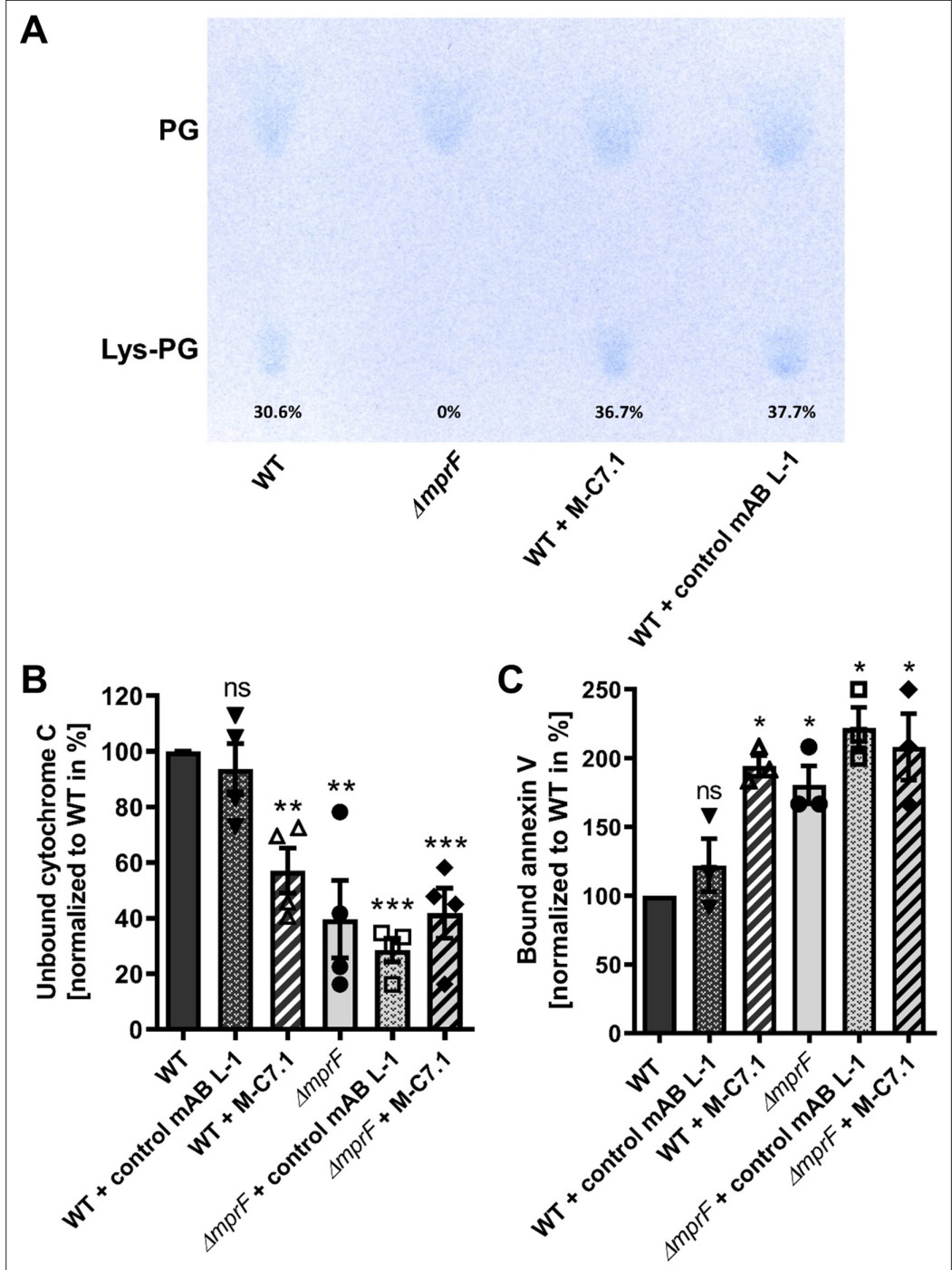

**Figure 4.** M-C7.1 inhibits the multiple peptide resistance factor (MprF) lysyl-phosphatidylglycerol (LysPG) flippase but not the LysPG synthase. (**A**) Detection of phospholipids from *S. aureus* SA113Δ*mprF* and wild-type (WT) treated or not treated with 100 µg/ml M-C7.1 or the isotype control monoclonal antibody (mAB) L-1. Polar lipids were separated by thin layer chromatography (TLC) and stained with the phosphate group-specific dye molybdenum blue to detect the well-documented phosphatidylglycerol (PG) and LysPG pattern of *S. aureus* WT and *mprF* deletion mutant (***Slavetinsky et al., 2012***). Percentages of LysPG in relation to total phospholipid content are given below LysPG spots. (**B**) The repulsion of positively charged cytochrome C corresponds to MprF LysPG synthase plus flippase activity while the synthase activity alone does not affect repulsion. To assess MprF flippase efficiency, unbound cytochrome C in the supernatant was quantified photometrically after incubation with the *S. aureus* SA113 WT without pretreatment, or with pretreated with M-C7.1 or the isotype control mAB L-1 (WT set to 100%). SA113Δ*mprF* with or without mAB pretreatment served as positive control. The means + standard error

*Figure 4 continued on next page*

*Figure 4 continued*

of the mean (SEM) of results from three biological replicates are shown. (**C**) Annexin V binding to *S. aureus* SA113 WT compared to incubation with M-C7.1 or the isotype control mAB L-1 was quantified by measuring cell-bound annexin V by fluorescence-activated cell sorting (FACS) and untreated WT samples were set to 100%. SA113Δ*mprF* with and without mAB incubation served as positive control. Data are expressed as % of untreated WT cells. The means + SEM of results from three biological replicates are shown in (B) and (C). Significant differences compared to WT samples were calculated by Student's paired *t*-test (*p < 0.05; **p < 0.01; ***p < 0.0001).

membrane surfaces to accomplish LysPG translocation (*Figure 6*), or it could be in a unique position of the large MprF protein complex that allows access from both sides. This protein part might represent the center of the MprF flippase establishing the previously reported MprF-dependent distribution of charged LysPG in the inner and outer leaflet of the cytoplasmic membrane of *S. aureus* (*Ernst et al., 2009*; *Figure 6*). While this manuscript was in revision, the three-dimensional structure of the MprF protein of *Rhizobium tropici* was reported. It demonstrates that the region corresponding to the *S. aureus* MprF loop 7 is part of a gate between two large protein cavities, which are accessible from the outer and inner surface of the cytoplasmic membrane and allow for the passage of LysPG molecules between the two membrane leaflets (*Song et al., 2021*). Song et al. suggest a mechanistic model where LysPG binding in the inner protein cavity of the MprF flippase results in a transient channel formation/gate opening allowing LysPG to diffuse into the outer protein cavity which would put the M-C7.1-binding site in the center of the flipping process. This model would explain why M-C7.1 binding to MprF seems to require the live *S. aureus* cell to achieve structural accessibility to MprF. Moreover, Song et al. confirmed that MprF forms homodimers and larger oligomers. These structural data support our functional data and provide molecular explanations.

Interestingly, in living staphylococcal cells, M-C7.1 bound to putative higher-order MprF multimers as shown via blue native Western blotting suggesting that such multimers might represent the active lipid translocating state of MprF. Replacement of conserved amino acids in the M-C7.1-targeted protein part has been shown to inhibit the flippase function of MprF (*Ernst et al., 2015*), which underscores the crucial function of this domain. Notably, one of the conserved and essential amino acids in TMS 7, D254, is negatively charged. It is tempting to speculate that D254 may interact with the positively charged head group of LysPG during the translocation process in the *S. aureus* MprF. Mutations on loop 7 and adjacent TMS have recently been found to cause specific resistance to the

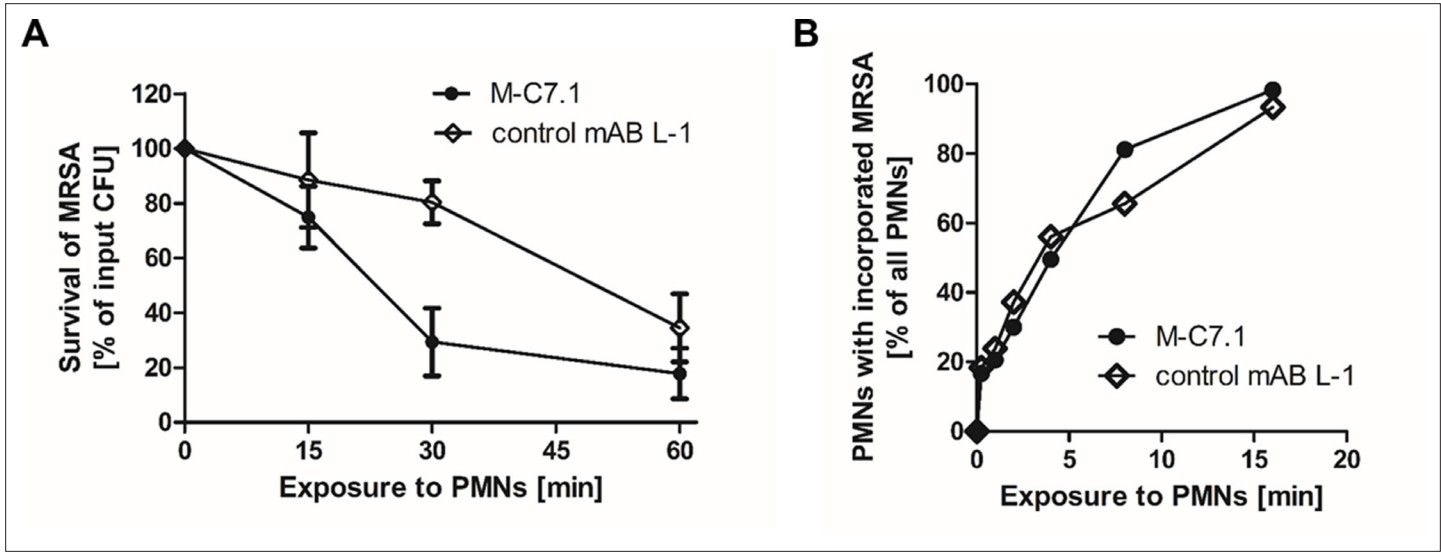

**Figure 5.** M-C7.1 supports *S. aureus* clearance by isolated human polymorphonuclear leukocytes (PMNs). (**A**) Kinetics of killing of community-associated methicillin-resistant *S. aureus* (CA-MRSA) strain USA300 wild-type (WT) treated with M-C7.1 compared to isotype control monoclonal antibody (mAB) L-1 by freshly isolated human PMNs. Viable bacteria (colony-forming units, CFU) after incubation with PMNs are shown as percentage of initial CFU counts. The means + standard error of the mean (SEM) of results from three biological replicates are shown. (**B**) Kinetics of phagocytosis of USA300 WT treated with M-C7.1 compared to isotype control mAB L-1 by freshly isolated human PMNs. Percentages of PMNs bearing fluorescein-5-isothiocyanate (FITC)-labeled USA300 are given. Means of three counts from a representative experiment are shown.

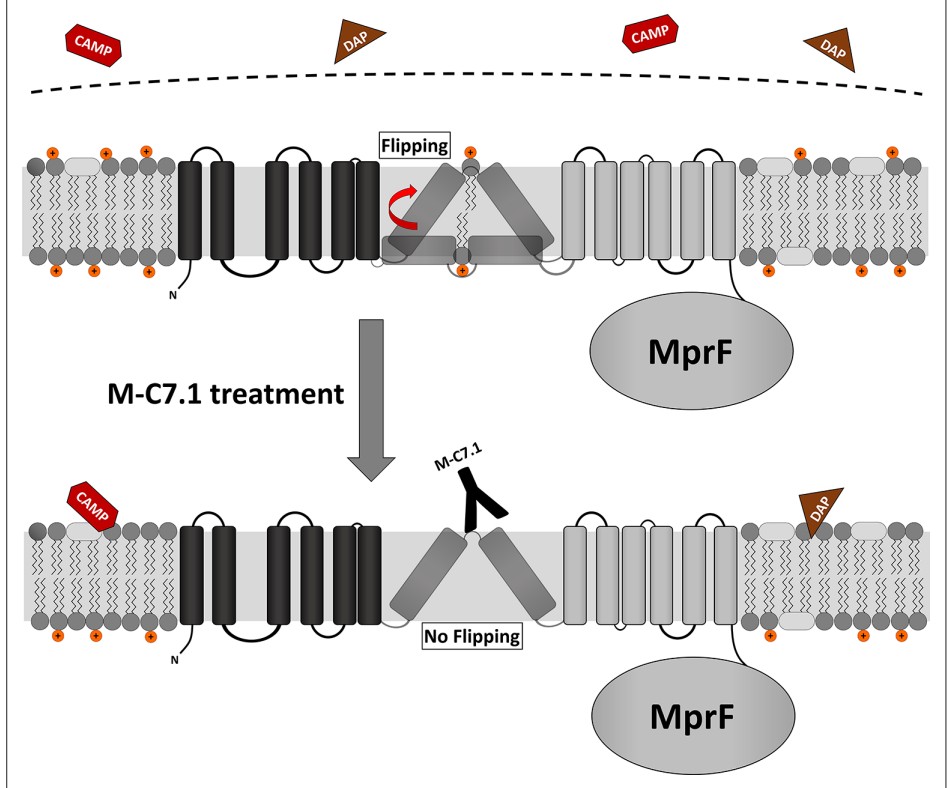

**Figure 6.** Proposed model for multiple peptide resistance factor (MprF) inhibition by M-C7.1. Flipping of positively charged lysyl-phosphatidylglycerol (LysPG) probably by transmembrane segment (TMS) 7 and 8 of MprF results in a more positively charged staphylococcal membrane, which is better able to repulse cationic antimicrobial peptides (CAMPs) or daptomycin (DAP). Binding of M-C7.1 to MprF loop 7 blocks the flippase, which results in a more negatively charged staphylococcal membrane and subsequently in an increased *S. aureus* membrane disruption by CAMPs and daptomycin.

structurally related lipopeptide antibiotics daptomycin and friulimicin B but not to other antimicrobials (*Ernst et al., 2018*). Indirect evidence has suggested that these mutations allow the flippase to translocate these lipopeptide antibiotics instead of or together with LysPG (*Ernst et al., 2018*), which is in agreement with a direct role of loop 7 in the translocation process. A mAB-binding loop 13 also sensitized *S. aureus* to nisin in a similar but less pronounced way as M-C7.1 suggesting that this loop may also have a critical role for MprF activity.

With the severe shortage of new antibiotic target and drug candidates, crucial cellular machineries conferring fitness benefits during infection rather than accomplishing essential cellular functions should be considered as targets for the development of new therapeutics. The phospholipid synthase and flippase MprF may become a role model for further targeting of bacterial defense mechanisms for such antifitness or antivirulence drugs. Compared to other potential targets, MprF has the advantage that essential parts of it are exposed at the external surface of the cytoplasmic membrane. Moreover, MprF is found in many other bacterial pathogens for which similar therapeutic mABs could be developed (*Ernst et al., 2009*). On the other hand, only a few amino acids in loop 7 are conserved in other bacteria, which would limit the impact of loop-7-directed mABs on other microbiome members thereby minimizing the resistance selection pressure and the risk of dysbiosis.

## Materials and methods
### Bacterial strains, maintenance, and mutagenesis of mprF

We used commonly used strains, the methicillin-susceptible laboratory strain *S. aureus* SA113 (ATCC 35556), methicillin-resistant clinical clone *S. aureus* USA300, and the SA113 *mprF* knockout derivative

SA113Δ*mprF*, which has been described recently (**Peschel et al., 1999**; **Wang et al., 2007**; **Supplementary file 1b**).

For the construction of a protein A mutant (Δ*spa*) the *E. coli/S. aureus* shuttle vector pKOR1 (**Bae and Schneewind, 2006**) was used, which allows allelic replacement with inducible counterselection in staphylococci. Flanking regions of *spa* were amplified from chromosomal DNA of *S. aureus* COL with primer pairs Spa-del_attB1 (ggggacaagtttgtacaaaaaagcaggc**caatattccatggtccagaact**; bold: spa sequence) and Spa-del for BglII (*gtcgagatc*tataaaaacaaacaatacacaacg, restriction site italic) as well as Spa-del_attB2 (ggggaccactttgtacaagaaagctggg**atcagcaagaaaacacacttcc**; bold: spa flanking sequence) and Spa-del rev BglII (*aaaagatct*aacgaattatgtattgcaata, restriction site italic). Both PCR products were digested with BglII and subsequently ligated. Without further purification, the ligation product was mixed with equimolar amounts of pKOR1 and in vitro recombination was performed with BP clonase Mix (Invitrogen) according to the manufacturer's instructions. The recombination mixture was transferred to chemically competent *E. coli* DH5α and isolated plasmids from the resulting transformants were analyzed by restriction digest. The correct plasmid was isolated from *E. coli* DH5α and used to electroporate competent *S. aureus* RN4220 from which it was again isolated and transferred into *S. aureus* SA113 by electroporation. Allelic replacement was essentially conducted as previously described (**Bae and Schneewind, 2006**) and resulting deletion mutants were confirmed by PCR.

The *S. aureus* SA113 *spa mprF* double mutant (SA113Δ*spa*Δ*mprF*) was constructed by transducing the gene deletion cassette of the *S. aureus* SA113 *mprF* deletion mutant (SA113Δ*mprF*) to the markerless *spa*-deficient *S. aureus* SA113 mutant (SA113Δ*spa*) using standard transduction protocols. The resulting *S. aureus* strain SA113Δ*spa*Δ*mprF* was identified by screening for erythromycin resistance conferred by the gene deletion cassette and confirmed by PCR of the deleted genome section. Bacteria were maintained on tryptic soy agar plates.

Cysteine substitutions in *mprF* were accomplished by site-directed mutagenesis in *E. coli* using *E. coli/S. aureus* shuttle vector pRB474 bearing *mprF*, using the QuickChange II Site-Directed Mutagenesis Kit (Agilent), as described recently (**Ernst et al., 2018**). Mutated *mprF* derivatives in pRB474*mprF* were transferred into SA113Δ*mprF* and *mprF* expression was mediated by the plasmid-encoded constitutive *Bacillus subtilis* promoter *vegII*. 10 μg/ml chloramphenicol served for maintenance in all plasmid-based studies. A99, T263, and T480 were chosen for exchange against a cysteine residue because of prediction of a weak effect on protein structure in an analysis of functional changes given a single point mutation according to https://predictprotein.org (**Yachdav et al., 2014**).

Plasmids and primers used in this study are given in **Supplementary file 1c, d**, respectively. Unless otherwise stated, bacteria were cultivated in Mueller-Hinton Broth (MHB, Sigma-Aldrich) with appropriate antibiotics for all experiments.

## Antigen selection and antibody production

MprF-derived peptides of interest (**Figure 1** and **Supplementary file 1a**) were custom synthesized by JPT Peptide Technologies GmbH (Berlin). An N-terminal and a C-terminal cysteine were added to enable cyclization. Biotin was coupled to the peptide via a 4,7,10-trioxa-1,13-tridecanediamine succinic acid (Ttds) linker (**Ieronymaki et al., 2017**). Cyclization and coupling of linker and biotin were performed by JPT Peptide Technologies.

Recombinant antibodies were generated from the HuCAL PLATINUM library, as described recently, by three iterative rounds of panning on the antigen peptides in solution and antigen–antibody-phage complexes were captured with streptavidin-coated beads (Dynal M-280) (**Prassler et al., 2011**; **Tiller et al., 2013**). Fab-encoding inserts of the selected HuCAL PLATINUM phagemids were cloned and expressed in *E. coli* TG1 F cells, purified chromatographically by IMAC (BioRad) and subcloned in an IgG1 expression vector system to obtain full-length IgG1s by expression in eukaryotic HKB11 cells, as described recently (**Prassler et al., 2011**). Identity has been identified by STR profiling. Regular mycoplasma contamination testing was performed throughout all mAB productions and were negative. IgGs were purified by protein A affinity chromatography (MabSelect SURE, GE Healthcare) (**Prassler et al., 2011**).

The humanized antibody MOR03207 (**Neuber et al., 2014**) directed against chicken lysozyme serves as isotype control and was called L-1 in this manuscript. The IgG MOR13182 generated with peptide QSIDTNSHQDHTEDVEKDQSE derived from the *S. aureus* surface protein elastin-binding

protein of *S. aureus* (EbpS) served as positive control for whole-cell ELISAs and was called E-1 in this manuscript.

## Peptide and whole-cell ELISAs of MprF-specific mABs

Overnight *S. aureus* SA113Δ*spa* or SA113Δ*spa*Δ*mprF* cultures grown in MHB were adjusted to $OD_{600}$ 0.1 in fresh MHB and grown to $OD_{600}$ of at least 1.0. Cells were harvested, adjusted to $OD_{600}$ 0.5 with 0.9% NaCl, and used to coat 96 well NUNC Maxisorp Immuno Plates for 1 hr (50 µl/well). After three washing steps with Tris-buffered saline (TBS; 50 mM Tris, 150 mM NaCl in 800 ml of $H_2O$, pH 7.4) containing 0.05% Tween 20 (TBS-T), the cells were blocked with PBS containing 1× ROTIBlock (Carl Roth) for 1 hr. Primary antibodies were added after three washing steps with TBS-T at an indicated final concentration of 1, 10, or 100 nM and incubated for 1 hr. After three more washing steps with TBS-T, antihuman IgG conjugated to alkaline phosphatase (Sigma-Aldrich A8542) diluted 1:10,000 in TBS-T was applied. Finally, after three washing steps with 2× TBS-T and 1× TBS, the cells were incubated with 100 µl *p*-nitrophenyl phosphate solution as advised by the manufacturer (Sigma-Aldrich N1891). Absorbance was measured at 405 nm.

## Detection of antibody binding to MprF by blue native Western blotting

To detect binding of the anti-MprF antibody to MprF we performed blue native polyacrylamide gel electrophoresis (BN-PAGE) as previously described (**Ernst et al., 2015**) with slight modifications. Briefly, 20 ml of *S. aureus* cells grown in MHB with anti-MprF antibodies (100 µg/ml) to $OD_{600}$ 1 were incubated with 750 µl of lysis buffer (100 mM ethylenediaminetetraacetic acid (EDTA) [pH 8.0], 1 mM $MgCl_2$, 5 µg/ml lysostaphin, 10 µg/ml DNase, proteinase inhibitor [cocktail set III from Calbiochem] diluted 1:100 in PBS) for 30 min at 37°C and homogenized three times with 500 µl zirconia-silica beads (0.1 mm diameter from Carl-Roth) using a FastPrep 24 homogenizer (MP Biomedicals) for 30 s at a speed of 6.5 m/s. After removing the beads, cell lysate was centrifuged 20 min at 14,000 rpm and 4°C to sediment cell debris and supernatant was transferred to microcentrifuge polypropylene tubes (Beckman Coulter) and cytoplasmic membranes were precipitated by ultracentrifugation for 45 min at 55,000 rpm and 4°C (Beckman Coulter rotor TLA 55). Membrane fractions resuspended in resuspension buffer (750 mM aminocaproic acid, 50 mM Bis-Tris [pH 7.0]) were incubated with the MprF- or lysozyme-specific humanized IgG mAB and/or a GFP-specific rabbit IgG mAB (Invitrogen) for 30 min shaking at 37°C. Dodecyl maltoside was added to a final concentration of 1% for 1 hr at 4°C to solubilize MprF–antibody complexes. Insoluble material was removed by ultracentrifugation for 30 min at 40,000 rpm and 4°C. 20 µl supernatant was mixed 1:10 with 10× BN loading dye (5% [wt/vol] Serva Blue G, 250 mM aminocaproic acid, 50% glycerol) and run on a Novex NativePAGE 4%–16% Bis-Tris gel (Invitrogen). Separated proteins were transferred to a polyvinyl difluoride membrane and detected via either, goat antihuman IgG DyLight 700 (Pierce) or goat antirabbit IgG DyLight 800 (Pierce) as secondary antibodies in an Odyssey imaging system from LI-COR.

## SCAM to localize MprF loops in inner and/or outer membrane leaflet

SCAM was adapted for use in *S. aureus* from the protocol recently established for *E. coli* (**Bogdanov et al., 2005**). Variants of *S. aureus* SA113Δ*mprF* were expressing cysteine-deficient and/or -altered MprF derivatives bearing a FLAG tag at the C-terminus via the plasmid pRB474. Bacteria were grown at 37°C to an $OD_{600}$ 0.7–0.8 in 100 ml TSB, split into two equal aliquots, and harvested. Pellets were resuspended in a mixture of buffer A (100 mM HEPES, 250 mM sucrose, 25 mM $MgCl_2$, 0.1 mM KCl [pH 7.5]) supplemented with 1 mM $MgSO_4$, 1 mM EDTA, 2.1 µg lysostaphin, 0.7 µg DNase, and 1% of a protease inhibitor cocktail (Roche) and incubated for 45 min at 37°C. For staining of cysteine residues in the outer leaflet of the cytoplasmic membrane, cells of the first aliquot were treated with *N*\*-(3-maleimidylpropionyl)-biocytin (MPB, Thermo Scientific) at a final concentration of 107 µM for 20 min on ice. In the second aliquot, external cysteine residues were blocked with 107 µM (AMS Thermo Scientific) for 20 min on ice, bacterial cells were subsequently disrupted in a bead mill as described above and internal cysteines were labeled with MPB for 5 min. MPB labeling was quenched in both aliquots by the addition of 21 mM β-mercaptoethanol. Cell debris was removed by several centrifugation and washing steps, membrane fractions were collected via ultracentrifugation at 38,000 × *g*, and labeled membrane samples were stored at −80°C.

For protein enrichment and precipitation, labeled membrane samples were thawed on ice and suspended in 20 mM β-mercaptoethanol containing buffer A. Proteins were solubilized by the addition of an equal volume of solubilization buffer (50 mM Tris–HCl [pH 9], 1 mM EDTA, 2% SDS) and vigorous vortexing for 30 min at 4°C, followed by an incubation step of 30 min at 37°C and vortexing for 30 min at 4°C. After solubilization, one and a half volumes of immunoprecipitation buffer 1 (50 mM Tris–HCl [pH 9], 150 mM NaCl, 1 mM EDTA, 2% Thesit, 0.4% SDS [pH 8.1]) was added and samples were incubated for 2.5 hr with magnetic FLAGbeads (Sigma-Aldrich) in a rotation wheel at 4°C. After several washing steps with immunoprecipitation buffer one and immunoprecipitation buffer 2 (50 mM Tris–HCl [pH 9], 1 M NaCl, 1 mM EDTA, 2% Thesit, 0.4% SDS [pH 8.1]), FLAG-tagged MprF was eluted by the addition of 0.1 M glycine–HCl (pH 2.2) and neutralized by adding Tris–HCl (pH 9).

For detection of cysteine-labeled MprF, samples were analyzed by denaturing SDS–PAGE and Western blotting. To this end, 12 µl samples were mixed with 4× Laemmli sample buffer (BioRad), loaded onto a 10% polyacrylamide gel, and separated via electrophoresis. Proteins were transferred onto a polyvinylidene difluoride (PVDF) membrane (Immobilon-PSQ PVDF membrane, Merck) by semi-dry turbo Western blot procedure (Trans-Blot Turbo Transfer System, BioRad). FLAG-tagged MprF was detected both by a mouse anti-FLAG primary antibody (Sigma-Aldrich) and goat antimouse secondary antibody (LI-COR) at 700 nm, while MBP-labeled cysteine residues were detected with streptavidin DyLight conjugate (Thermo Scientific) at 800 nm in an Odyssey imaging system from LI-COR.

## Determination of susceptibility to antimicrobial agents

Overnight cultures of *S. aureus* SA113Δ*spa* or USA300 WT were diluted in fresh MHB and adjusted to $OD_{600}$ 0.25 (~$1.5 \times 10^7$ cells). Antibodies were adjusted to a concentration of 1 mg/ml and 10 µl per well of a 96-well plate were added to 90 µl of the adjusted cell suspension (final antibody concentration: 100 µg/ml). Cells were grown in the presence of the isotype control antibody L-1 or with the respective anti-MprF antibodies. After 3 hr of incubation at 37°C under shaking, optical density was determined, and cells were adjusted to $OD_{600}$ 0.025 in 500 µl cold PBS. 80 µl of the adjusted cell suspension were mixed with 20 µl of antimicrobial substances to final concentrations of 22.5 µg/ml nisin for SA113Δ*spa* or of 5 µg/ml nisin for USA300, 1.5 µg/ml daptomycin for USA300, 11 µg/ml daptomycin for DAP-R CA-MRSA strain 703, and 45 µg/ml LL-37 for USA300 or 20 µl PBS as control. After incubation for 2 hr under shaking at 37°C, the cell suspensions were diluted 1:200 and 100 µl of each duplicate was plated in triplicates on TSB agar plates to obtain a representative value of bacterial survival.

The capacity of M-C7.1 to reduce daptomycin MICs of *S. aureus* was analyzed by pretreatment of SA113 WT, SA113Δ*mprF*, and DAP-R CA-MRSA strain 703 with either PBS (1:10 diluted), control mAB L-1 (100 µg/ml), or M-C7.1 (100 µg/ml) in MHB, cultivation for 3 hr at 37°C under agitation. Subsequently, bacterial suspensions were adjusted to $OD_{600}$ 0.05, and daptomycin MIC test strips (Liofilchem) were applied according to the manufacturer's advice.

To analyze the capacity of mAB M-C7.1 to inhibit *S. aureus* growth, overnight cultures of CA-MRSA strain USA300 grown in MHB medium were diluted to $OD_{600}$ 0.01 with fresh MHB medium containing 1 µM of antibodies and 4 µg/ml nisin (Sigma-Aldrich) or PBS as a control in 96-well plates. Plates were incubated for 24 hr at 37°C under shaking conditions using a Bioscreen C (Oy Growth Curves Ab Ltd). Optical density at 600 nm was determined and compared to growth without nisin.

## Determination of in vitro evolution of resistance to (combined daptomycin) M-C7.1 treatment in *S. aureus*

*S. aureus* USA300 was inoculated in a 48-well microtiter plate (Nunclon Delta surface, Thermo Scientific Nunc) with 500 µl MHB supplemented with 50 mg/l CaCl₂ at 37°C and shaking. It was combined with either (1) only subinhibitory concentrations of daptomycin (0.5 µg/ml), (2) the same daptomycin concentrations plus 10 nM or 100 nM M-C7.1, (3) only 10 nM or 100 nM M-C7.1 without daptomycin, or (4) no additional supplements. Cultures were passaged every 24 hr for 6 days in total. Control wells were included to check for sterility. To check for potential contaminations, 5 µl of each passage was streaked on Tryptic Soy Agar to detect single colonies. To screen for potential mutations in *mprF* resulting from respective selection pressure, genomic DNA was isolated from each passage using NucleoSpin Microbial DNA Kit (Marchery-Nagel) according to the manufacturer's instructions. *MprF* was

amplified with the primers MprF_USA300_fw and MprF_USA300_rev. Amplified *mprF* was sequenced by Sanger sequencing using primers MprF_USA300_fw, MprF_USA300_600, MprF_USA300_1200, MprF_USA300_1800, MprF_USA300_2200, MprF_USA300_800rev and MprF_USA300_rev by Eurofins Genomics Europe and analyzed for point mutations.

## Isolation and quantification of polar lipids

*S. aureus* cultures (1 ml in MHB) were grown to the exponential phase ($OD_{600}$ 1) in the presence of antibodies (final concentration: 100 µg/ml) as described above for the determination of susceptibility to antimicrobials. Polar lipids were extracted using the Bligh–Dyer method (*Bligh and Dyer, 1959*), with chloroform/methanol/sodium acetate buffer (20 mM) (1:1:1, by vol), vacuum-dried, and resuspended in chloroform/methanol (2:1, by vol). Extracts were filled into a 100 µl Hamilton syringe and spotted onto silica gel 60 F254 high-performance thin-layer chromatography plates (Merck) with a Linomat five sample application unit (Camag) and run in a developing chamber ADC 2 (Camag) with a running solution composed of chloroform–methanol–water (65:25:4, by vol). Phosphate group-containing lipids were detected by molybdenum blue staining and phospholipids were quantified in relation to total phospholipid content by determining lipid spot intensities densitometrically with ImageJ (http://rsbweb.nih.gov/ij/docs/guide/index.html) as described recently (*Slavetinsky et al., 2012*).

## Determination of *S. aureus* cell surface charge

For analysis of the cytochrome C repulsion capacity of *S. aureus*, bacterial cells were grown in the presence of antibodies (100 µg/ml) as described above for the analysis of antimicrobial susceptibility. Bacteria were adjusted to $OD_{600}$ 1 in 1 ml PBS, pelleted, resuspended in 100 µl 0.05 mg/ml cytochrome C (Sigma-Aldrich) followed by incubation at 37°C under shaking. The cells were then pelleted by centrifugation and absorption of the supernatant containing unbound cytochrome C was determined by measuring absorbance at 410 nm.

To determine annexin V binding to *S. aureus*, bacterial cells were grown in the presence of antibodies (100 µg/ml) as described above at 37°C with shaking for 3 hr, harvested, washed twice in PBS buffer, and resuspended in PBS containing $CaCl_2$ to $OD_{600}$ 0.5 in 1 ml. Bacteria were gently mixed with 5 µl of allophycocyanin-labeled annexin V (Thermo Scientific) and incubated at room temperature for 15 min in the dark, as described recently (*Ernst et al., 2018*). At least 50,000 bacterial cells per antibody were analyzed by flow cytometry to quantify surface-bound fluorophore (FL-4).

## Phagocytosis and killing of *S. aureus* by human (PMNs)

Human PMNs are the major human phagocytes to counteract bacterial infection and form the largest subgroup (~95%) of PMNs (*Spaan et al., 2013*). PMNs were isolated from fresh human blood of healthy volunteers by standard Ficoll/Histopaque gradient centrifugation as described recently (*Hanzelmann et al., 2016*). Cells were resuspended in Hanks' balanced salt solution (HBSS) containing 0.05% human serum albumin (HBSS-HSA; HBSS with 0.05% Albiomin, Biotest AG).

CA-MRSA strain USA300 was prepared for PMN experiments as described for the determination of susceptibility to antimicrobials, grown in the presence of antibodies (100 µg/ml) at 37°C under shaking for 3 hr, harvested, washed twice in HBSS, and adjusted to a density of $5 \times 10^7$ CFU/ml. Pooled serum from healthy human volunteers (blood bank of the University Hospital Tübingen) was added to a final concentration of 5% and bacteria were opsonized for 10 min at 37°C, as described recently (*Peschel et al., 2001*). Prewarmed bacteria and PMNs were mixed to final concentrations of $5 \times 10^6$ CFU/ml and $2.5 \times 10^6$ PMNs/ml in flat-bottom 96-well plates together with antibodies (final concentration 100 µg/ml). Samples of 50 µl were shaken at 37°C.

For analysis of the PMN killing capacity, incubation was stopped at different time points by the addition of 100-fold volumes of ice-cold distilled water to disrupt the PMNs. Triplets of appropriate sample volumes were spread on LB agar plates and colonies were counted after 24 hr incubation at 37°C. Numbers of live bacteria did not change during the 60 min incubation period in the absence of PMNs compared to the initial bacteria counts.

For phagocytosis studies, overnight cultures of bacteria were labeled with 0.1 mg/ml fluorescein-5-isothiocyanate (FITC) at 37°C for 1 hr, as described previously (*Peschel et al., 2001*). After washing with PBS, the bacteria were resuspended in HBSS-HSA, adjusted to $5 \times 10^6$ CFU/ml, mixed with PMNs ($2.5 \times 10^6$ PMNs/ml), and opsonized as described above (shaking at 37°C). Incubation was stopped

by addition of 100 µl ice-cold 1% paraformaldehyde. The percentage of PMNs bearing FITC-labeled bacteria was determined by flow cytometric analysis of 20,000 cells.

## Statistics

Statistical analyses were performed with the Prism 5.0 package (GraphPad Software) and group differences were analyzed for significance with the two-tailed Student's *t*-test. A p value of ≤0.05 was considered statistically significant.

## Acknowledgements

This work was financed by Grants from the Deutsche Forschungsgemeinschaft (DFG) (TRR34, SFB766, TRR156, TRR261, and GRK1708) to AP; and the German Center of Infection Research (DZIF) to AP. CJS was supported by the intramural Experimental Medicine program of the Medical Faculty at the University of Tübingen and received grants from the DZIF and the DFG Cluster of Excellence EXC2124 'Controlling Microbes to Fight Infection' (CMFI). The authors acknowledge infrastructural support by the cluster of Excellence EXC2124 CMFI, project ID 390838134.

## Additional information

### Competing interests

Alexandra Kraus, Michael Tesar, Christoph M Ernst, Andreas Peschel: Antibodies disclosed in the manuscript are part of patent "Anti-staphylococcal antibodies" (US9873733B2 / EP2935324B1), assigned to the Universitaetsklinikum Tuebingen and MorphoSys AG. The authors have declared that no further conflict of interest exists. The other authors declare that no competing interests exist.

### Funding

| Funder | Grant reference number | Author |
|---|---|---|
| Deutsche Forschungsgemeinschaft | EXC-2124/project ID 390838134 | Christoph J Slavetinsky Andreas Peschel |
| Deutsches Zentrum für Infektionsforschung | TTU 08.806 | Andreas Peschel Christoph J Slavetinsky |
| Deutsche Forschungsgemeinschaft | SFB 766/1-3 TP A08 | Andreas Peschel |
| Deutsche Forschungsgemeinschaft | TRR34 | Andreas Peschel |
| Deutsche Forschungsgemeinschaft | TRR156 | Andreas Peschel |
| Deutsche Forschungsgemeinschaft | TRR261 | Andreas Peschel |
| Deutsche Forschungsgemeinschaft | GRK1708 | Andreas Peschel |
| Universitätsklinikum Tübingen | | Christoph J Slavetinsky |
| MorphoSys | | Christoph M Ernst |

The funders had no role in study design, data collection, and interpretation, or the decision to submit the work for publication.

### Author contributions

Christoph J Slavetinsky, Conceptualization, Data curation, Formal analysis, Funding acquisition, Investigation, Methodology, Project administration, Supervision, Validation, Visualization, Writing – original draft, Writing – review and editing; Janna N Hauser, Formal analysis, Investigation, Methodology, Writing – review and editing; Cordula Gekeler, André Geyer, Doris Heilingbrunner, Investigation,

Methodology; Jessica Slavetinsky, Investigation, Methodology, Visualization; Alexandra Kraus, Conceptualization, Data curation, Investigation, Methodology, Supervision, Writing – review and editing; Samuel Wagner, Concept and analysis of Blue-Native Western blotting, Conceptualization, Data curation, Methodology, Writing – review and editing; Michael Tesar, Conceptualization, Data curation, Methodology, Writing – review and editing; Bernhard Krismer, Investigation, Methodology, Writing – review and editing; Sebastian Kuhn, Data curation, Methodology; Christoph M Ernst, Conceptualization, Data curation, Formal analysis, Investigation, Methodology, Project administration, Supervision, Validation, Writing – review and editing; Andreas Peschel, Conceptualization, Formal analysis, Funding acquisition, Project administration, Supervision, Writing – original draft, Writing – review and editing

### Author ORCIDs
Christoph J Slavetinsky ⓘ http://orcid.org/0000-0001-5576-5906
Samuel Wagner ⓘ http://orcid.org/0000-0003-1808-3556
Christoph M Ernst ⓘ http://orcid.org/0000-0002-5575-1325
Andreas Peschel ⓘ http://orcid.org/0000-0002-3209-8626

### Decision letter and Author response
Decision letter https://doi.org/10.7554/eLife.66376.sa1
Author response https://doi.org/10.7554/eLife.66376.sa2

## Additional files

### Supplementary files
• Supplementary file 1. Multiple peptide resistance factor (MprF)-directed antibodies, bacterial strains, plasmids, and primers.
 (a) MprF-directed antibodies and its target peptides. (b) Bacterial strains used in this study. (c) Plasmids used in this study. (d) Primers used in this study.

• Transparent reporting form

### Data availability
All data generated or analysed during this study are included in the manuscript and supporting files.

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
