## [Editor Report]

This study uses an innovative anti-virulence approach based on monoclonal antibodies that target the *Staphylococcus aureus* lipid flippase involved in tolerance to cationic peptides. The authors show that this strategy re-sensitizes antibiotic-resistant *S. aureus* and serves as a proof of principle for anti-virulence approaches to target bacterial infections.

---

## [Decision Letter]

**Decision letter after peer review:**

Thank you for submitting your article "Sensitizing *Staphylococcus aureus* to antibacterial host defense by decoding and blocking the lipid flippase MprF" for consideration by *eLife*. Your article has been reviewed by 3 peer reviewers, and the evaluation has been overseen by a Reviewing Editor and Bavesh Kana as the Senior Editor. The following individuals involved in review of your submission have agreed to reveal their identity: Dorte Frees (Reviewer #2); Gabriella Marincola (Reviewer #3).

Essential revisions:

This study would be of interest to readers who specialise in infectious diseases research and treatment. The authors developed a monoclonal antibody that sensitises Methicillin-resistant *Staphylococcus aureus* to host peptides and antibiotics. Treatment with the antibody decreased bacterial survival in phagocytes. Overall, the antibody could be used as potential new therapeutic, diminishing the severity of *S. aureus*-associated disease. At this stage, the in vivo efficacy remains unknown.

1. A more thorough assessment of CAMP sensitization to show a dose response is needed to fully support the conclusions (i.e., checkerboard analysis with increasing concentrations of antibody and CAMPs).

2. The authors should address the potential for resistance to develop. Since mprF mutations confer resistance to CAMPs, did the authors consider assessing the propensity of the target to mutate in the presence of their monoclonal antibody?

3. The anti-virulence scope, as indicated by the title and the introduction, is not aligned with the actual content of the Results section and needs to be improved. The in vivo potential of anti-virulence strategies is evident throughout the introduction, but is only preliminarily address in the results. In contrast, a large part of Results section (lines 146-214) investigates the position of one specific loop of the MprF protein.

4. Figure 1A and throughout the manuscript. Although the antigens are predicted to be exposed to the extracellular site of the membrane they would not be accessible to antibodies because of the cell wall, as was shown for LTA. This polymer despite being exposed to the exterior of the membrane is not accessible to antibodies unless the cell wall is degraded. This crucial point needs to be discussed. (Reichmann NT et al., Differential localization of LTA synthesis proteins and their interaction with the cell division machinery in *Staphylococcus aureus*. Mol Microbiol. 2014 Apr;92(2):273-86. Doi: 10.1111/mmi.12551. PMID: 24533796; PMCID: PMC4065355.)

5. Figure 1B, would the IgG binding Sbi impact the result? Could the observed differences between wt and mutant be indirect (the changed membrane composition in the mprF cells may impact broadly on secretion of proteins, hence, unspecifically changing the surface antigenicity of the cells). I acknowledge that a humanized isotype mAB was used as a control, but to rule out the indirect effect, I think it would be better to use an antibody targeting a *S. aureus* surface molecule as a control.

6. Line 171-173, “the 900-kDa band of MprF-GFP was detected by both M-C7.1 and anti-GFP confirming the identity of MprF”. This conclusion is not well supported: the anti-GFP detects a smear of proteins in this area in both Figure 2A and Figure S4, and seems to have some specificity for a protein complex around 480 kDa that was, however, not recognized by the C7.1 antibody. How can this be explained and how does this affect the conclusions?

7. How does the unspecific binding observed for the C7.1 antibody (to the 300 kDa band) affect the results shown in Figure 1B and Figure 3? In Figure 3, how does an mprF mutant survive the same treatment? It could also be relevant to test the effect in a strain background with gain of function mutations in mprF.

8. Untreated wt and mprF strains should be included in Figure 4BC to fully evaluate this result.

9. A mprF negative mutant and preferentially also a mprF gain of function mutant should be included as controls in Figure 5.

10. Line 235-240. The impact of the work would increase if the authors determine and show the Daptomyicin MIC of the DAP-R strain with or without M-C7.1 (as control they could use the strains from Suppl. Figure S3: ∆mprF complemented with empty plasmid (low MIC) or with intact MprF (high MIC)).

11. Line 129-135 and Supplemental Figure 2. In the Figure one can see that the peptide corresponding to loop 1 binds not just to its own mAb (M-C1) but also to the ones directed against loop 7 and 9 (M-C7.2 and M-C9.2). This is the reason why the authors did not work further with these two mAbs. They write in line 134 “Antibodies M-C1, M-C7.1, M-C7.3, M-C9.1, M-C13.1, and M-C13.2 bound significantly stronger to MprF….” (Figure 1 B). However, all of them do (including the two which show less selectivity in the supplemental figure). I find this to be a bit confusing for the reader.

12. Line 154-186. Although the M-C7.1 antibody does seem to bind MprF (and I am aware of the challenge involved in blotting a blue native gel and obtaining sharp visible bands), it is very difficult to see and follow bands in figure 2A. Arrows which point to the putative bands the authors are referring to in the text (i.e. the 250, 500 and 900 kDa bands), would be extremely helpful for the reader. Also, the gel image in supplemental figure 4 is much more intuitive. Maybe the authors want to consider using these instead of 2A?

13. Line 196-200. The authors write that cysteines in MprF do not have critical functions and indeed the mutant complemented with the cysteine-deficient MprF or with the other version of MprF showed an increased MIC towards Dapto (Supplemental Figure 3). However, it is also clear that they do NOT do it to the same extent as a wild type MprF (Supplemental Figure 3 black column). It would be important to discuss this result. (In this figure, it would be easier for the reader to add a ∆mprF label below the graph, to immediately understand that it is the mutant complemented with various plasmids).

---

## [Author Response]

This study would be of interest to readers who specialise in infectious diseases research and treatment. The authors developed a monoclonal antibody that sensitises Methicillin-resistant Staphylococcus aureus to host peptides and antibiotics. Treatment with the antibody decreased bacterial survival in phagocytes. Overall, the antibody could be used as potential new therapeutic, diminishing the severity of *S. aureus*-associated disease. At this stage, the in vivo efficacy remains unknown.1. A more thorough assessment of CAMP sensitization to show a dose response is needed to fully support the conclusions (i.e., checkerboard analysis with increasing concentrations of antibody and CAMPs).

We agree that a dose response analysis would be helpful to support the conclusions. We include now a new Figure 3 —figure supplement 1 demonstrating that M-C7.1 synergizes with nisin in a dose-dependent fashion. These data are mentioned in the Results section (lines 229-231):

“mAB M-C7.1 caused the strongest sensitization (Figure 3A) and it synergized with nisin in a dose-dependent fashion (Figure 3 —figure supplement 1).”

2. The authors should address the potential for resistance to develop. Since mprF mutations confer resistance to CAMPs, did the authors consider assessing the propensity of the target to mutate in the presence of their monoclonal antibody?

We agree that mutations in *mprF* that could lead to spontaneous resistance to M-C7.1 are a potential risk to address in the manuscript. We performed a new experiment, based on passaging *S. aureus* over six days in media with subinhibitory daptomycin concentrations and with two M-C7.1 concentrations but no point mutations in *mprF* occurred, suggesting that M-C7.1 is not prone to rapid resistance induction. These data are mentioned in the Results section (lines 246-249):

“When USA300 WT was passaged for several days through media with M-C7.1 at 10 nM or 100 nM and with or without subinhibitory daptomycin (0.5 µg/ml), no point mutations in mprF were found suggesting that the MprF segment targeted by M-C7.1 is not prone to quickly occurring escape mutations..”

3. The anti-virulence scope, as indicated by the title and the introduction, is not aligned with the actual content of the Results section and needs to be improved. The in vivo potential of anti-virulence strategies is evident throughout the introduction, but is only preliminarily address in the results. In contrast, a large part of Results section (lines 146-214) investigates the position of one specific loop of the MprF protein.

We changed the title as advised to:

“Sensitizing *Staphylococcus aureus* to antibacterial agents by decoding and blocking the lipid flippase MprF”

The text addressing anti-virulence therapies was strongly reduced in the introduction as suggested. Moreover, we added a sentence on the importance of basic functional studies for paving the way for anti-virulence strategies. The modified section in the Introduction reads now (lines 72-74):

“Moreover, in-depth molecular studies are necessary to devise most promising targets for mABs and elucidate if and how mAB binding could disable pathogens to colonize and infect humans”

4. Figure 1A and throughout the manuscript. Although the antigens are predicted to be exposed to the extracellular site of the membrane they would not be accessible to antibodies because of the cell wall, as was shown for LTA. This polymer despite being exposed to the exterior of the membrane is not accessible to antibodies unless the cell wall is degraded. This crucial point needs to be discussed. (Reichmann NT et al., Differential localization of LTA synthesis proteins and their interaction with the cell division machinery in *Staphylococcus aureus*. Mol Microbiol. 2014 Apr;92(2):273-86. doi: 10.1111/mmi.12551. PMID: 24533796; PMCID: PMC4065355.)

It is true that the capacities of antibodies to reach the cytoplasmic membrane of *S. aureus* is controversial. Nevertheless, several studies in addition to ours have demonstrated that such an interaction is possible, probably via large cavities and pores in the peptidoglycan, as recently revealed by Simon Foster’s lab (Pasquina-Lemonche L *et al.*, The architecture of the Gram-positive bacterial cell wall. Nature 582:294-297). This issue is addressed now in the Discussion section (lines 297-302):

“The capacity of antibodies to traverse the cell wall and reach the cytoplasmic membrane of *S. aureus* has remained controversial. However, other labs’ and our findings demonstrate that specific antibodies can reach membrane-associated epitopes in *S. aureus*. A recent atomic-force microscopy-based study revealed large, irregular pores in the cell wall of *S. aureus*, which might allow large proteins such as antibodies to reach the cytoplasmic membrane.”

5. Figure 1B, would the IgG binding Sbi impact the result? Could the observed differences between wt and mutant be indirect (the changed membrane composition in the mprF cells may impact broadly on secretion of proteins, hence, unspecifically changing the surface antigenicity of the cells). I acknowledge that a humanized isotype mAB was used as a control, but to rule out the indirect effect, I think it would be better to use an antibody targeting a *S. aureus* surface molecule as a control.

We are grateful for this suggestion, and we use now monoclonal antibody E-1 binding the surface-exposed elastin-binding protein of *S. aureus* (EbpS) now as an additional control (changed Figure 1B). The new data further support our findings and conclusion. They are mentioned in the Results section (lines 137-139):

“An additional mAB directed against the *S. aureus* surface protein EbpS served as positive control, showing equal affinity towards SA113*Δspa* and SA113*ΔspaΔmprF* (Figure 1B).”

6. line 171-173, "the 900-kDa band of MprF-GFP was detected by both M-C7.1 and anti-GFP confirming the identity of MprF". This conclusion is not well supported: the anti-GFP detects a smear of proteins in this area in both Figure 2A and Figure S4, and seems to have some specificity for a protein complex around 480 kDa that was, however, not recognized by the C7.1 antibody. How can this be explained and how does this affect the conclusions?

We agree with the reviewer that this part needs better explanation. The experiment shown in Figure 2A is meant to show the non-covalent interaction of MprF or MprF-GFP oligomers with M-C7.1. Since SDS-PAGE would disrupt these interactions, we had to use blue-native PAGE, which yields less defined protein bands. Nevertheless, blue-native PAGE is a valuable and widely accepted technique and the identity of the band at around 900 kDa is clear as it is detects the MprF(-GFP)-M-C7.1 complex by anti-human secondary antibody and is absent in the *mprF* mutant. This band could also be detected with the anti-GFP antibody, albeit with lower efficiency compared to detection with M-C7.1. It is possible that the MprF-GFP-M-C7.1 complex shields the GFP tag from binding by the GFP antibody to some degree. On the other hand, M-C7.1 (used as primary antibody as well) seems to bind the non-complexed MprF-GFP oligomer less efficiently than the M-C7.1-complexed MprF-GFP oligomer, which may explain that the MprF-GFP band at ca. 480 kDa (described as 500 kDa in the manuscript) is hardly visible when detected with M-C7.1. We assume that M-C7.1 binds MprF in live *S. aureus* cells preferentially during the process of LysPG flipping which is supported by the recently solved structure of *Rhizobium tropici* MprF, published while this manuscript was in revision (Song, D., H. Jiao, and Z. Liu, Phospholipid translocation captured in a bifunctional membrane protein MprF. Nat Commun, 2021. 12(1): p. 2927). It shows that the M-C7.1 target sequence is in the putative active center of the flippase. This center forms a barrier / gate for LysPG and the authors speculate that it opens to a channel during the flipping process. Such a conformational accessibility would explain why only the MprF-M-C7.1 complex is detected (directly by the anti-human secondary antibody) and non-M-C7.1 bound MprF oligomers are hard to detect with M-C7.1 (as primary antibody).

However, the specificity of protein bands that appeared upon complexation of MprF-GFP oligomers with M-C7.1 is clearly demonstrated by the absence of these bands in the *mprF* mutant and in samples complexed with control antibody L1. We moved the previous Figure S4 to the main manuscript as new Figure 2A and deleted and modified the corresponding text to explain the findings better. The modified text in the Results section reads (lines 182-185):

“The fact, that M-C7.1 used as primary antibody was not able to detect 250 and 500-kDa bands of non-complexed MprF proteins while the MprF-M-C7.1 complex can directly be detected via secondary antibody, suggests that M-C7.1 binding only occurs in the living staphylococcal cell.”

The modified text in the Discussion section reads (lines 322-324):

“This model would explain why M-C7.1 binding to MprF seems to require the live *S. aureus* cell to achieve structural accessibility to MprF.”

7. How does the unspecific binding observed for the C7.1 antibody (to the 300 kDa band) affect the results shown in Figure 1B and Figure 3?

We find unspecific binding of M-C7.1 only to isolated proteins in blue-native PAGE (Figure 2A) but not with intact *S. aureus* cells (Figure 1B). M-C7.1 may therefore unspecifically bind an antigen that is not accessible in intact cells, e.g. a cytoplasmic protein.

In Figure 3, how does an mprF mutant survive the same treatment? It could also be relevant to test the effect in a strain background with gain of function mutations in mprF.

We include a new Figure 3F, which shows inhibition of *S. aureus* WT, *mprF* knockout mutant, and gain-of-function mutant strain 703 by daptomycin with or without M-C7.1. It confirms that the *mprF* mutant is much more susceptible to daptomycin and that M-C7.1 does not further lower its susceptibility. Strikingly, it also demonstrates that both WT and strain 703 with an amino acid exchange in a protein region of the MprF loop targeted by M-C7.1 is also sensitized to daptomycin by M-C7.1. These data are mentioned in the Results section (lines 241-242):

“Of note, M-C7.1 could reduce daptomycin minimal inhibitory concentration (MIC) of both, *S. aureus* SA113 WT and the DAP-R strain 703 (Figure 3F),…”

8. Untreated wt and mprF strains should be included in Figure 4BC to fully evaluate this result.

The requested controls were added to new Figure 4B and 4C, which confirm our findings. These experiments show that the WT and *mprF* mutant strains bind similar amounts of cytochrome c or annexin V when incubated without or with control mAB L-1 while M-C7.1 treatment alters the binding capacities of the WT. These data are mentioned in the Results section (lines 260-263):

“Treatment of SA113 WT with M-C7.1 at concentrations that increased the susceptibility to nisin led to a significant increase in the capacity to bind cytochrome C or annexin V compared to SA113 WT without treatment or treated with isotype control mAB L-1…”

9. A mprF negative mutant and preferentially also a mprF gain of function mutant should be included as controls in Figure 5.

We have demonstrated that *S. aureus mprF* mutants are equally well phagocytosed but more efficiently inactivated by human neutrophils in two previous publications (Peschel et al., J Exp Med 2001 193:1067-76; Kristian et al., Infect Immun 2003 71:546-9). That M-C7.1 does not further affect the susceptibility of an *S. aureus mprF* mutant to cationic antimicrobial peptides is shown in new Figure 3F. We hope therefore, it will not be necessary to include more controls in Figure 5, which would require high amounts of M-C7.1, most of which has been used up now for the other experiments. We addressed this point in the Results section (line 270-272):

“Accordingly*, S. aureus mprF* mutants are more susceptible to PMN killing than the parental strains while their opsonization and phagocytosis be PMNs remains unaltered.”

10. Line 235-240. The impact of the work would increase if the authors determine and show the Daptomyicin MIC of the DAP-R strain with or without M-C7.1 (as control they could use the strains from Suppl. Figure S3: ∆mprF complemented with empty plasmid (low MIC) or with intact MprF (high MIC)).

We agree that such experiments would significantly increase the impact of the work and are very grateful for this suggestion. The requested experiments are shown now in new Figure 3F, see comments above. These finding support the conclusions of our study.

11. Line 129-135 and Supplemental Figure 2. In the Figure one can see that the peptide corresponding to loop 1 binds not just to its own mAb (M-C1) but also to the ones directed against loop 7 and 9 (M-C7.2 and M-C9.2). This is the reason why the authors did not work further with these two mAbs. They write in line 134 "Antibodies M-C1, M-C7.1, M-C7.3, M-C9.1, M-C13.1, and M-C13.2 bound significantly stronger to MprF…." (Figure 1 B). However, all of them do (including the two which show less selectivity in the supplemental figure). I find this to be a bit confusing for the reader.

We agree that this point is somewhat confusing, and we added new text to clarify (lines 128-129):

“Peptide 1 bound IgGs developed against different antigen peptides indicating that it may bind antibodies with only low selectivity.”

Further, we deleted “and selectivity” in the following sentence to point out that antibodies were selected based on affinity.

12. Line 154-186. Although the M-C7.1 antibody does seem to bind MprF (and I am aware of the challenge involved in blotting a blue native gel and obtaining sharp visible bands), it is very difficult to see and follow bands in figure 2A. Arrows which point to the putative bands the authors are referring to in the text (i.e. the 250, 500 and 900 kDa bands), would be extremely helpful for the reader. Also, the gel image in supplemental figure 4 is much more intuitive. Maybe the authors want to consider using these instead of 2A?

We are grateful for this suggestion and exchanged the Western blot images from previous Figure 2A and Figure 2 —figure supplement 2. Moreover, the MprF-specific bands at 250, 500 and 900 kDa are now indicated in new Figure 2A. Figure 2 —figure supplement 2 and Figure 2A were changed. Markers in new Figure 2 —figure supplement 2 were included as well.

13. Line 196-200. The authors write that cysteines in MprF do not have critical functions and indeed the mutant complemented with the cysteine-deficient MprF or with the other version of MprF showed an increased MIC towards Dapto (Supplemental Figure 3). However, it is also clear that they do NOT do it to the same extent as a wild type MprF (Supplemental Figure 3 black column). It would be important to discuss this result. (In this figure, it would be easier for the reader to add a ∆mprF label below the graph, to immediately understand that it is the mutant complemented with various plasmids).

Again, we are grateful for this helpful suggestion. The Figure was modified accordingly, and the findings better described and discussed (lines 201-203):

“However, the mutated proteins conferred lower resistance levels than the wild-type protein, presumably because of less efficient protein folding or stability.”